# DENETHOR: The DynamicEarthNET dataset for Harmonized, inter-Operable, analysis-Ready, daily crop monitoring from space

**Lukas Kondmann**[1,2], **Aysim Toker**[3], **Marc Rußwurm**[1,7], **Andrés Camero**[2],
**Devis Peressutti**[4], **Grega Milcinski**[4], **Nicolas Longépé**[5], **Pierre-Philippe Mathieu**[5],
**Timothy Davis**[6], **Giovanni Marchisio**[6], **Laura Leal-Taixé**[3], **Xiao Xiang Zhu**[1,2*]

[1]Data Science in Earth Observation, Technical University of Munich (TUM)
[2]Earth Observation Center, German Aerospace Center (DLR)
[3]Dynamic Vision and Learning Group, Technical University of Munich (TUM)
[4]Sinergise
[5]Φ-Lab, European Space Agency (ESA)
[6]Planet Labs
[7]Environmental Computational Science and Earth Observation Laboratory (ECEO),
École Polytechnique Fédérale de Lausanne (EPFL)

## Abstract

Recent advances in remote sensing products allow near-real time monitoring of the Earth's surface. Despite the increasing availability of near-daily time series of satellite imagery, there has been little exploration of deep learning methods to utilize the unprecedented temporal density of observations. This is particularly interesting in crop monitoring where time series remote sensing data has been used frequently to exploit phenological differences of crops in the growing cycle over time. In this work, we present **DENETHOR**: The **D**ynamic**E**arth**NET**[2] dataset for **H**armonized, inter-**O**perable, analysis-**R**eady, daily crop monitoring from space. Our dataset contains daily, analysis-ready Planet Fusion data together with Sentinel-1 radar and Sentinel-2 optical time series for crop type classification in Northern Germany. Our baseline experiments underline that incorporating the available spatial and temporal information fully may not be straightforward and could require the design of tailored architectures. The dataset presents two main challenges to the community: Exploit the temporal dimension for improved crop classification and ensure that models can handle a domain shift to a different year.[3]

## 1 Introduction

Remote sensing is entering a new era of time series analysis. A growing number of commercial and public satellites take the pulse of our planet in unprecedented frequency and resolution. Modern satellites reimage the Earth in ever-shorter time intervals generating petabytes of data every year [49]. Additionally, the open data policy of the Landsat program in the USA [46] and the Copernicus program by the European Space Agency (ESA) [1] have enabled the use of Earth observation (EO) data for many applications.

---

*Corresponding author: xiaoxiang.zhu@dlr.de.

[2]DynamicEarthNET is the larger project under which our institutions collaborate to make multi-temporal Earth observation (EO) data more accessible.

[3]All model implementations and data are available at https://github.com/lukaskondmann/DENETHOR

One task at the heart of remote sensing efforts is vegetation monitoring. Particularly for the study of vegetation, access to frequent time series data is essential for accurate and timely monitoring of forests [47] and agricultural activities [26, 29]. The Sentinel-2 mission, which collects multispectral data at up to 10 m resolution at least every 5 days, has become particularly popular for crop type classification since its launch in 2015 [27, 33, 34, 38].

Recent advances in remote sensing have, however, made it possible to go beyond the spatial and temporal resolution of Sentinel-2. The Planet Fusion product, for instance, provides daily coverage of the Earth in 3m resolution and is part of a larger group of next-generation EO products that deliver analysis-ready data in dense time intervals. In the future, high temporal acquisition frequencies with near-daily intervals may become the norm in vegetation monitoring. This allows observing the growing cycle of crops in near real-time which provides significant potential for this field. However, the current methods in crop type classification are not designed to make use of daily temporal imagery, particularly in combination with high spatial resolution.

Therefore, in this paper, we present the dataset DENETHOR which provides the first opportunity to explore analysis-ready, daily data for crop type mapping. We provide a combination of harmonized, declouded, daily Planet Fusion data at 3m resolution together with Sentinel-1 and 2 time series for high-quality field boundaries and crop ids in Northern Germany. Train and test tiles are spatially separated and taken from different years to encourage out-of-year generalization.

We explore three types of benchmark methods on the dataset with the daily data: At first, we take the mean per pixel per field over time as input to a temporal encoder which discards spatial information but scales well [36]. Second, we include a spatial encoder in combination with the temporal encoder. Third, we follow [38] by randomly sampling pixels from a parcel as input to a temporal self-attention model. We compare these approaches to a random forest baseline with handcrafted spectral features from Sentinel-1 and 2.

Our experiments with the daily time series provide a starting point for future methodological approaches and underline that current methods may not yet be able to use the full potential of available information. We find that simply including a spatial encoder in addition to the temporal backbone does not improve performance compared to a simple mean of pixels per field. Second, many competitive deep learning models tested struggle to surpass the random forest baseline based on Sentinel-1 and 2 on our test set. One of the only approaches which manage this is based on pixel-set encoding and temporal self-attention [12, 38] with a high score of about 2/3 in accuracy. Finally, we underline that the performance drop in out-of-year evaluation from 2018 and 2019 can be substantial and amounts to 12 percentage points in accuracy on DENETHOR. To summarize, our contribution is threefold:

- We introduce the first publicly available benchmark dataset that includes daily, analysis-ready remote sensing data. With this, we aim to incentivize the community to study when and how this data source can be useful for crop type monitoring.

- Our experiments outline that mapping crops from daily imagery may require new methods since exploiting the full potential of the inputs seems to be a hurdle for many baseline models. Therefore, the dataset provides a challenging opportunity for the machine learning community with a novel type of input data.

- We emphasize the necessity of crop type models to be robust to domain shift not only along the spatial but also along the temporal dimension. This may be an underestimated problem in practice which we outline in our baseline results.

## 2   Related Work

Crop type classification is a special case of land cover classification in agricultural monitoring where field boundaries are typically assumed to be known. Because of its necessity for yield prediction and food security estimations, the task has received considerable attention in the past. Especially, multitemporal EO data has been a primary source for crop type classification for decades [26, 29]. Initially, however, the temporal scale of information provided a computational challenge. Therefore, early methods relied mostly on feature extraction from the time series [35]. Popular approaches include the computation of vegetation indices [5, 6, 11, 13, 28] that are often combined with Random Forest Classifiers [42] or Support Vector Machines [8, 20, 48]. Further, Dynamic Time Warping (DTW) [25] has found many applications in phenological studies with time series EO data [2, 7, 24].

More recently, however, the rise of deep learning in artificial intelligence [21] has also fueled improvements in crop type classification. Recurrent neural networks such as LSTMs [15] are well-suited to capture the temporal dynamics of crop types in a satellite time series [33, 40, 34]. Conversely, convolutional approaches can exploit the spatial dependency of the data. Further, [27] introduce the use of temporal convolutional neural networks (TempCNNs) in crop type mapping where convolutions are applied also along the temporal dimension. Convolutional and recurrent approaches have also been combined to leverage spatial as well as temporal information [18].

Attention mechanisms can further improve upon the capabilities of recurrent models [43]. Self-attention can be particularly effective when applied to raw optical time series as attention mechanisms can distinguish between informative and cloudy images [35]. A temporal attention encoder with pixel-set encoders (PSE) is successfully applied to randomly sampled pixels of crop parcels in [38].

The majority of recent methodological developments are based on publicly available medium-resolution satellite data from Sentinel-2 (S2). S2 has a spatial resolution of up to 10m per pixel and collects 13 spectral bands. Its revisit time is 5 days meaning that any region on Earth with a size larger than 100km² will be reimaged at least every 5 days [9]. The S2 mission has driven substantial methodological progress in remote sensing, especially because of its open data policy [49]. Still, on average 55% of the earth's surface is covered by clouds [19]. This can decrease the temporal resolution in practice notably and impede the aim of missions, such as Copernicus, to provide frequent and reliable observations [10].

Recent advances in remote sensing technology have induced the next-generation of optical EO products with superior temporal coverage compared to S2. The increased temporal resolution can be especially helpful against clouds since it improves the chance of a cloud-free observation substantially. Commercial missions such as the Planetscope constellation achieve *near daily* revisit times. The spatial and temporal resolution of next-generation EO products holds great promise for a variety of applications in monitoring our planet. However, current methods in deep learning have not been designed to fully exploit the available temporal information at scale. This may be primarily an issue of available temporal resolution in current benchmark datasets. Benchmark datasets for crop type mapping are scarce in remote sensing and are mostly based on S2 data. The rarity of datasets may primarily be a result of the low availability of high-quality reference data at scale. Large-scale products of crop types such as the Cropland Data Layer (CDL) [4] in the US exist for some regions. However, the CDL is technically still a prediction and may not provide sufficient quality for benchmark purposes.

Table 1 presents an overview of available benchmark datasets for crop type classification from multi-temporal EO data. Two datasets based in Europe provide a large evaluation ground for newly developed crop type mapping models. At first, Breizhcrops [36] is based on S2 and reference data from almost 800,000 fields from the Brittany region in France. Similarly, the TimeSen2Crop [45] dataset covers a large fraction of fields in Austria with S2 input data at about 1,200,000 parcels. Satellite data is averaged at the field level which makes it possible to include a high number of fields. This averaging, however, discards a lot of spatial information per field, essentially reducing each field to one averaged pixel. Both of these datasets prioritize geographical size since there are natural limits to the temporal resolution through the S2 revisit rate and cloud obstruction.

Further, several datasets from Africa have been open-sourced as competitions through the work of the Radiant Earth Foundation together with Zindi Africa and local governments. The first dataset from Kenya was part of the challenge at the computer vision for agriculture workshop at the International Conference for Learning Representations (ICLR) in 2020. Similarly, datasets from Uganda, Rwanda and South Africa have been or are used in competitions to develop innovative methods for crop type mapping based on S2 (Uganda), aerial images (Rwanda) and a combination of Sentinel-1 and 2 (South Africa).

None of these datasets, however, include the opportunity to push the frontier of current models further by including next-generation EO data. Our dataset DENETHOR aims to change this by releasing daily, declouded and harmonized Planet Fusion data in combination with S1 and S2 inputs for 4,500 fields in Germany. With this, we aim to incentivize the remote sensing and the machine learning community to develop methods to improve current approaches based on rich data for a challenging and relevant problem. Naturally, the focus on temporal and spatial resolution of images comes with the necessity to restrict the spatial scale of the dataset to keep access to it democratic and feasible. Still, the daily data with 3m resolution is the main driver of dataset size which sums up to about

Table 1: Existing Datasets for Crop Type Classification. GSD = Ground Sampling Distance, RT = Revisit Time

|  | Inputs | GSD | RT | #Fields | Size[GB] |
|---|---|---|---|---|---|
| Breizhcrops (FR) [36] | S2 | 10m | 5 days | 768,000 | 17.4 |
| TimeSen2Crop (AUT) [45] | S2 | 10m | 5 days | 1,200,000 | 2.1 |
| CV4A Kenya [30] | S2 | 10m | 5 days | 4,700 | 3.5 |
| Crop Type Uganda [3] | S2 | 10m | 5 days | 52 | 59.4 |
| Crop Type Rwanda [32] | UAV | 3cm | Monthly | 2,611 | 26.9 |
| Spot the Crop Challenge (SA) [31] | S1+S2 | 10m | 5 days | 35,300 | 52.1 |
| DENETHOR (Ours) | PF+S2+S1 | 3m | Daily | 4,500 | 254.5 |

255GB. The inclusion of S1 and S2 enables users of the dataset to further explore multi-modal combinations of input data.

Our dataset is the first of its kind to publish a daily product for scientific development at the intercept of machine learning and remote sensing. This does not only hold in the context of crop type mapping but in EO in general to the best of our knowledge. DENETHOR will be released under a CC-BY license to encourage widespread use and adoption.

## 3 DENETHOR: Daily Time Series for Crop Type Classification

**Crop Type Classes.** Our dataset includes field boundaries and crop type information from Northern Germany. This data is collected as part of the Common Agricultural Policy of the European Union. Farmers self-report the crops they grow in their fields to receive subsidies. The data is not only geographically precise but also of high quality since a variety of checks via in-situ measurements or EO data can potentially expose cheating.

Given the high spatial and temporal resolution of our dataset, we restrict our focus to two tiles. Both tiles are identical in size with 24km × 24km. One tile is used for training and validation, the other for testing. For the training tile, we include field masks and crop types from the year 2018 together with the respective satellite imagery. Test evaluations are based on the 2019 data. With this, we aim to encourage methodological development that incorporates not only a spatial but also a potential temporal shift in the input data. We will further release the 2018 test tile data and 2019 training tile data for future ablation studies.

The raw crop information provides the fields in vector format with a crop id coded between 1-999. Fields with areas below 1000m² are excluded since they are often broken in shape and can not easily be incorporated. This affects around 1% of all fields. We aggregate the crop type into a limited set of high-level classes which is common practice in crop type mapping [36]. The nine classes with the respective number of fields in the training set in brackets are: Wheat (305), Rye (276), Barley (137), Oats (45), Corn (251), Oil Seeds (201), Root Crops (23), Meadows (954) and Forage Crops (339). The class imbalance provides a challenge for machine learning algorithms but it is representative of the geographic region and an imbalance is generally common in real-world crop type mapping tasks [36]. Crops that do not fit into these categories are rare in the reference data but in these instances, we remove the respective fields instead of collecting them in a tenth 'Other' class.

**Planet Fusion Imagery.** The main source of imagery is the Fusion Monitoring product[4] by Planet Labs, a commercial provider of high-resolution satellite imagery. It is based on the Planetscope constellation of Cubesats which collect images of the Earth from over 180 small satellites. The product has a spatial resolution of 3m and collects 4 spectral bands (RGB + Near-infrared (NIR)). Although Fusion imagery is primarily used for early crop detection, plant health monitoring and the classification of phenological plant cycles, it has several features that may make it promising for crop type classification. At first, it provides imagery in a unique daily time interval which allows studying the evolution of crops in unprecedented temporal density. Especially in combination with the high spatial resolution, this could enable classification methods to pick up on small, crop-specific details of the growing cycle.

---

[4]https://www.planet.com/pulse/planet-announces-powerful-new-products-at-planet-explore-2020/

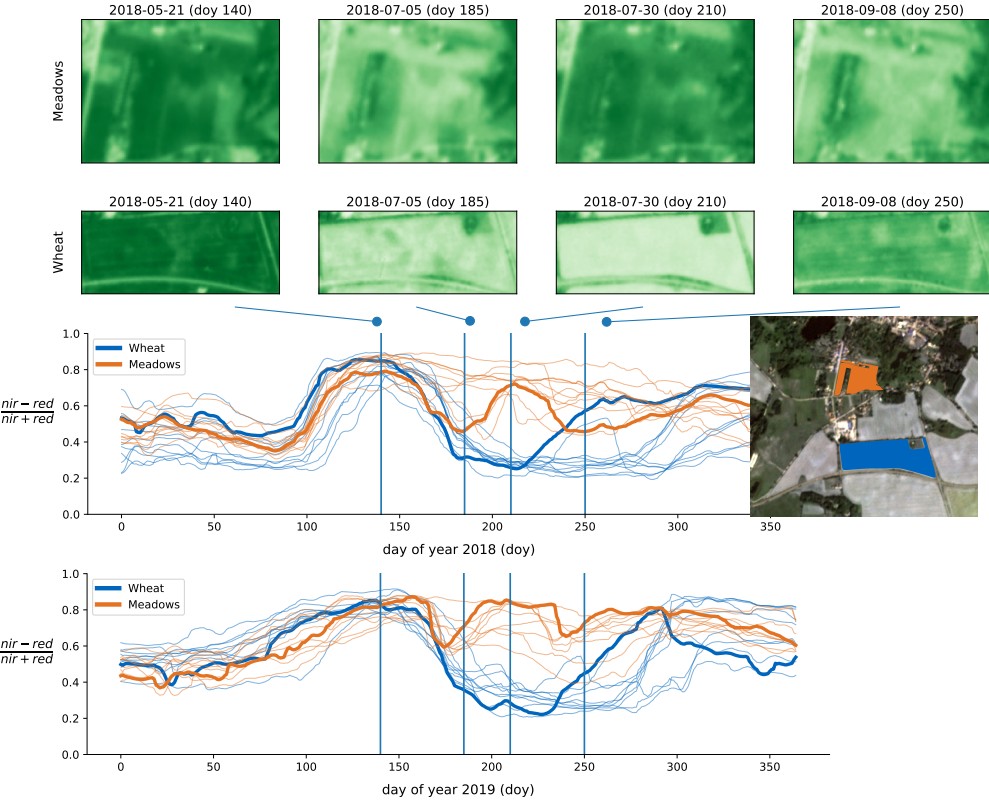

Figure 1: Examples of meadow and wheat parcels from the training dataset. The upper images show two parcels at four (of 365) acquisition times. The bottom plots shows the Normalized Difference Vegetation Index NDVI [41] averaged over all pixels of these parcels across the years 2018 (above) and 2019 (below). For reference, other fields meadow and wheat fields are plotted in thin lines which illustrates that wheat and meadow vary systematically between day 180 and 240 in both years. However, the vegetation activity varies notably between the years.

Second, it delivers a temporally consistent collection of images with removed clouds and shadows which in remote sensing is referred to as an analysis-ready (ARD) product. Potential gaps because of clouds are filled with different points in time. The data also includes quality assurance (QA) information that underlines from which source any pixel is taken from together with a confidence score if the observation is gap-filled. This may be particularly useful in combination with machine learning techniques since observation could be prioritized by their confidence measure.

Third, it is a Harmonized Landsat Sentinel-2 (HLS) time series.[5] To ensure interoperability with Landsat and Sentinel-2 data, the Fusion data is calibrated to the HLS spectrum which eliminates differences in the spectral signatures. These differences are subtle but important because they may, for example, lead to the fact that the red channel a Planetscope sensor collects is slightly different from the red band in Sentinel-2. The removal of these differences may also enable additional potential through data fusion. As Planet Fusion data is typically not freely available, our dataset provides an opportunity for the academic community to explore advantages and disadvantages of this data.

Figure 1 illustrates an example of what kind of signal multispectral time series data carry to map vegetation activity with two neighboring fields in our dataset. The two rows plot the normalized difference vegetation index (NDVI) around the meadow (top) and wheat (below) field for four points in time in a year. The 1D timelines at the bottom add the mean NDVI for the two selected parcels (thick lines) and other fields of the same crop (thin lines) in 2018 (above) and 2019 (below). After day 180 in the year, wheat has been harvested with a systematic decline in photosynthetic activity.

---

[5]https://earthdata.nasa.gov/esds/harmonized-landsat-sentinel-2

In contrast, meadows are still active with a high NDVI which - among many other features - can be captured by algorithms. Comparing the NDVI across 2018 and 2019 underlines the difficulty of out-of-year generalization in crop type mapping. While patterns show some similarity across years, there are also significant differences, particularly for meadows in the middle of the year.

**Sentinel data.** To combine and compare Planet and the publicly available Sentinel data, we include imagery from Sentinel-1 and Sentinel-2 to the train and test tile. While the spatial and temporal depth of S2 is comparably low, the combination of spectral depth (S2) with spatial and temporal depth (Planet Fusion) may provide additional opportunities for crop type mapping. The S2 data is downloaded from Sentinel Hub [6] at processing level L2A. No observations are filtered because of cloud cover (maxcc=1) to maximize temporal coverage. We provide 12 bands - all resampled at 10m resolution - together with the valid data mask, scene classification (SCL) band and the s2cloudless probability map (SCP).

Sentinel-1 (S1) is a radar-based sensor with a revisit time of 6 days [49]. We provide S1 Ground Range Detected (GRD) data with included orthorectification from Sentinel Hub. Orthorectification is the removal of terrain distortions in raw satellite images which stem from the fact that satellites rarely take images directly above the area of interest ('off-nadir'). We include both, vertical-vertical (VV) and vertical-horizontal (VH) polarization. The distinction stems from the fact that radar-based sensors collect information from electromagnetic waves that may be repolarized when they reach the surface. HH measures the share of waves that were emitted in vertical polarization and return to the sensor in the same polarization. Conversely, HV measures the fraction of the waves which are repolarized. Radar-based sensors are not obstructed by clouds and provide information about vegetation from a different perspective. Hence, a multi-modal approach based on optical and radar data could be informative of phenological trends on the ground.

**Possible tasks.** While the main focus of this study is crop type classification, the uniquely high cadence of the data sources can be used for continuous monitoring of crop vigor and precise identification of crop growth stages and drive progress in sustainable agricultural practices. It may also be particularly insightful to study approaches for early crop detection in the season when the full time series is not yet available. Further, instead of taking field boundaries as given, the direct segmentation of crops and fields [37] provides a higher level of difficulty for models. This could also be seen in the context of instance segmentation and connected to approaches to the MS coco challenge [22] where the different modalities of inputs may provide an interesting challenge. Further, arable land classification could be an intermediary step towards the direct segmentation of crops.

Beyond applications in crop monitoring, DENETHOR could provide validation exercises in super-resolution and declouding since we provide long time series of multi-modal data at different resolutions. For declouding, downsampled Planet Fusion data could be treated as the desired output with raw and cloudy sentinel inputs. On the other hand, cloud-free sentinel images could be used as input for a superresolution network that tries to upsample to 3m resolution. Finally, declouding and superresolution could also be combined in a single task.

## 4  Model Descriptions

**Deep learning models.** The listed models are evaluated only with Planet data as inputs. When training solely on S2 inputs, models did not converge. This is likely because S2 models may need a higher number of fields to enable training because of missing temporal and spatial resolution. The Planet multi-temporal satellite image sequence provides data at high spatial (3m) and temporal resolution (1 day). We benchmark three different ways of operationalizing the crop type mapping task from field boundaries and daily satellite images as visible in Figure 2.

Following a common practice in crop type mapping [27, 34], we consider a simple `pixel average` encoder (Figure 2 left) $f_{\text{spat}}(\mathbf{X}_t) = \frac{1}{hw} \sum_{r,c \in \text{mask}} \mathbf{x}_{\text{r,c,t}}$ over a field mask that averages each D-dimensional pixel $\mathbf{x}_{\text{r,c,t}}$ of a field into a $D$ dimensional vector. This discards spatial information for scalability.

Second, we include a rectangular image of each field (Figure 2 middle) at identical size (32x32). We sample down larger fields and zero pad smaller parcels. This preserves spatial and temporal information at the cost of increased input data of a factor of about 10³. We extract spatial and

---

[6]https://www.sentinel-hub.com

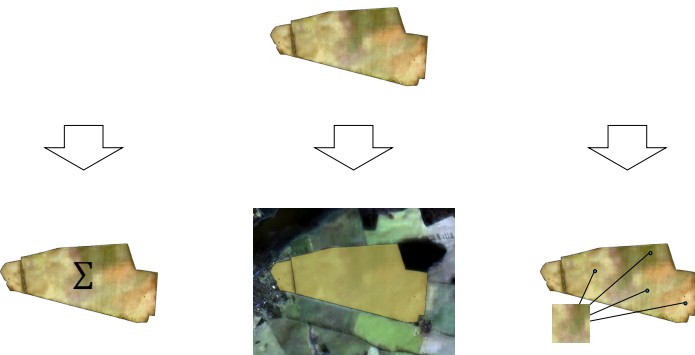

Figure 2: Three ways to operationalize the crop type mapping task with the field boundaries and satellite images as inputs (top): 1. Take the spatial mean of pixels within the field (left) 2. Take a rectangular image of identical size for all fields which can require padding or downsampling (middle) since fields vary in size. 3. Sample an identical number of random pixels from each field (right).

temporal features separately with two dedicated encoders. The spatial encoder $f_{\text{spat}} : \mathbb{R}^{h \times w \times D} \mapsto \mathbb{R}^H$ maps a single $D$-dimensional image of certain height $h$ and width $w$ into a $H$-dimensional feature vector while the temporal encoder $f_{\text{temp}} : \mathbb{R}^{T \times H} \mapsto [0,1]^C$ maps a sequence of $T$ $H$-dimensional feature vectors directly into a probability for one of C classes. The complete model $\{y_c\}_{c=1}^C = f_{\text{temp}}(\{f_{\text{spat}}(\mathbf{X}_t)\}_{t=1}^T)$ joins spatial encoder and temporal encoders such that the spatial encoder maps each image $\mathbf{X}_t$ into a H-dimensional representation that is then mapped to a class probability $y_c$ by the temporal encoder. In our case, $T = 365$ because we use daily observations and $C = 9$. $H$ depends on the model of the spatial encoder that is used.

For spatial encoders, we resort to standard, light-weight torchvision models [23], such as `mobilenetv3` [16], `squeezenet` [17], and `resnet18` [14]. We use imagenet pre-trained weights but replace the first layer to accommodate $D = 4$ input channels and use features before the classifier. The pixel average can be seen as a simple but scalable version of a spatial encoder.

For the temporal encoder, we utilize the provided implementations from the BreizhCrops [36] repository. We test the `TempCNN` [27] model that is based on three 1D-convolutions on the temporal dimension that are flattened and projected to class probabilities with a final dense layer. The 1D-Multi-Scale ResNet (`MSResNet`) model[7], is a variant of the CSI Net [44] for pose estimation and uses three CNN streams each with a different kernel size of 3, 5, 7. Features in the CNN streams are joined by residual skip connections. A fixed-size feature vector is obtained by global max pooling before the decision layer. We also compare to a recurrent neural network with multiple stacked LSTM layers[15], as explored early for crop type mapping by [34] and a `Transformer` Encoder [43], as tested in [35].

A third variant of spatial and temporal encoding has been proposed by [12, 38] where $f_{\text{spat}}$ is implemented as a `Pixel Set Encoder` (`Pse`). It transforms a set of random pixels (Figure 2 right) within a field parcel into a fixed representation by a pixel-wise MLP-based neural network with pooling. This strategy achieves good results when paired with a `Temporal Attention Encoder` (`Tae`) that is inspired by the self-attention-based Transformer architecture. This combination yields the `PseTae` [38] and a light-weight variant `PseLTae` [12].

**Data Fusion Baseline.** We compare the model accuracy with a random forest baseline on hand-designed features on openly available optical (Sentinel 2) and radar (Sentinel 1) data. For the 12-band Sentinel 2 and the 4-band RGB+NIR Planet data, we average all pixels of one field and consider only cloud-free observations which discards around 55% of S2 observations effectively cutting temporal resolution in half. Alongside the 12 or 4 spectral bands, we add the normalized difference vegetation index (NDVI) which is an index strongly related to vegetation. We calculate max, min, mean, median,

---

[7]https://github.com/geekfeiw/Multi-Scale-1D-ResNet

Table 2: Accuracy of Benchmark Models with Planet Fusion data on the 2019 test set trained with 2018 data by spatial (rows) and temporal encoder (columns). Naively using standard spatial encoders such as ResNet18, SqueezeNet or MobileNet is not sufficient to lift performance over a simple pixel average. PselTae is the best deep learning model but we observe a notable drop in performance of more than 20% compared to the validation set. The majority of this drop can be attributed to out-of-year prediction rather than the spatial shift.

| Spatial Encoder | Temporal Encoder | | | |
|---|---|---|---|---|
| | TempCNN [27] | MSResNet [44] | LSTM [34] | Transformer [35] |
| ResNet18 [14] | 52.22% | 49.53% | 44.64% | 43.61% |
| SqueezeNet [17] | 53.94% | 49.78% | 35.89% | 42.58% |
| MobileNetv3 [16] | 53.20% | 54.33% | 43.46% | 48.06% |
| Pixel Average [34] | 64.46% | 58.83% | 48.40% | 52.56% |
| Pixel-Set Encoding + Self-Attention | | | | |
| PselTae [12] | **67.25%** | | | |
| PseTae [38] | 64.95% | | | |
| Ablation Scores | | | | |
| PselTae (2018) | 78.77% | | | |
| PselTae (Val) | 88.02% | | | |

and standard deviation statistics on the resulting time series, as well as the date of max and min values for each of the 12 bands/4 bands plus NDVI index. For the Sentinel 1 radar data, we similarly average all pixels of one field at each time and calculate the same min, max, mean, median, std statistics on the vertical-vertical (VV) and vertical-horizontal (VH) polarisations. We additionally add the $\frac{VV}{VH}$ ratio as the third band. We calculate these features for ascending and descending orbits separately and concatenate the features from each orbit type.

## 5  Benchmark Results

All temporal encoder models were trained with their BreizhCrops [36] defaults and all spatial encoders are initialized with pretrained imagenet weigths from torchvision. We train with cross-entropy loss until convergence which typically occurs between epoch 50-100. Among the models, PselTae trained fastest with about 6 min per epoch with a batch size of 64 on a Nvidia GeForce GTX 1060.

Table 2 presents the accuracy test scores of our benchmarked approaches with the daily Planet data on the field level. The spatial encoder is given in the rows and the respective temporal encoder in the columns. Peak performances in this comparison group are reached by convolutional approaches with pixel average encoders with an accuracy of 64.46% for TempCNN and 58.83% for MSResNet. This stands in contrast to the results of [35] where self-attention outperforms convolutional approaches as temporal encoder. The notable difference is in the input data: When cloudy and raw Sentinel-2 data is used, convolutions may struggle to identify the relevant observations. In well-prepared, declouded images, temporal convolutions seem to excel and may hence be better suited for our dataset. All tested temporal encoders perform best with a simple pixel average as a spatial encoder. In alternatives to pixel averages, the choice of the spatial encoder seems to only make a marginal difference. This underlines that just including a deep-learning based spatial encoder does not work off-the-shelf and this kind of data may require more tailored approaches.

The highest score is reached by the light version of pixel-set encoding and self-attention (PselTae) with an accuracy of 67.25%. Given the validation accuracy of PselTae of 88.02%, this score is surprisingly low with a drop of over 20 percentage points (p.p.) between validation and test set. To split up this difference into a spatial and temporal shift, we evaluate PselTae as well on the test set in 2018 which results in 78.77% accuracy. Therefore, the spatial shift to a new tile accounts for about 9 p.p. and the temporal shift for 12 p.p. which is about 60% of the drop in total. Since accuracy scores may hide effects of class imbalance we also report macro-averaged F1 scores in the appendix in Table A1 which leaves the main impressions of Table 2 unchanged.

Table 3: Accuracy of different modalities with hand-designed features and a random forest classifier on the 2019 test set trained with 2018 data. Features are composed of 7 statistics (min, max, argmax, argmin, mean, median, std), for each band. Our Sentinel 2 data has 12 spectral bands plus a normalized red/near infrared ratio (7*(12+1)), Planet has 4 bands plus normalized red/near infrared ratio. Sentinel 1 has two bands plus the band ratio.

| Data Type | # Features | Accuracy | Macro F1-Score |
|---|---|---|---|
| Sentinel 1 (S1) | 42 | 0.58 | 0.43 |
| Sentinel 2 (S2) | 91 | 0.59 | 0.42 |
| Planet (PL) | 35 | 0.37 | 0.12 |
| S1 + S2 | 133 | 0.62 | **0.46** |
| S1 + PL | 77 | 0.60 | 0.42 |
| S2 + PL | 126 | 0.59 | 0.41 |
| S1 + S2 + PL | 168 | **0.63** | **0.46** |

Table 3 provides the scores of basic fusion experiments with random forests on the 2019 test set. S1 and S2 with hand-crafted features on their own reach accuracies of 58% and 59% respectively and 62% combined which surpasses all deep-learning models but PseLTae, PseTae and TempCNN with pixel average. The Random Forest approach does not work as well with the Planet data which is not surprising because it is tailored specifically for spectral rather than temporal or spatial depth. Beyond spectral features, textural features extracted from 3m data would likely contribute significantly to the differentiation of vegetation with Planet imagery. These textural features were not exploited in the current study but they could be highly complementary to the higher S2 spectral coverage.

The addition of Planet Fusion features to S1, S2 or to both adds some information that can be used by the random forest model but performance improves only marginally - if at all - in comparison to the models without PL. While the best deep learning models can surpass the performance of the Sentinel fusion baseline, this is not the case for most models implemented. Even if they do, the gap is fairly narrow at 2-5 p.p. accuracy. Extracting more advanced features from the Planet Fusion data with deep learning is therefore a promising route. Currently, however, it seems there may still be methodological potential in exploiting these novel kinds of inputs effectively at scale.

# 6 Discussion

The Planet Fusion data is a uniquely rich data source in the spatial and temporal dimension. However, our benchmark experiments suggest that our deep-learning baseline approaches may not be ideal to deal with the combination of high temporal and spatial resolution. The development of tailored architectures is opened as a challenge to the community to fully exploit the available information. One shortcoming of the tested models is that directly including spatial encoders before the temporal encoder makes performance worse compared to a simple spatial pixel average. PselTae/PseTae with Planet Fusion data are the best performing models but only reach an accuracy of about two-thirds. Therefore, much potential may be in improving current deep learning methods for daily input data. One promising route might be to experiment with pixel-set encoders with temporal convolutions since they were superior to attention networks as temporal encoders in our dataset. Likely, performance from daily data could also be improved notably with the inclusion of a larger geographic area as the models encounter this type of data for the first time. Nevertheless, a large geographic scale may not always be available in practice which underlines the necessity to develop approaches that can also learn from smaller regions and adds to the items that make our dataset challenging.

Further, we find a significant performance drop of 21 percentage points between our validation data in 2018 taken from the training tile and the 2019 test tile. We show that in our case about 60% of this decrease can be attributed to the temporal shift of just one year which is about 12 percentage points in accuracy. This is large in magnitude and documents a challenge of crop type mapping in practice. Since the weather and growing cycles can vary notably from year to year, out-of-year generalization provides a challenge. The size of the performance drop shows that this could be underestimated in real-world applications of crop type mapping since the potential magnitude of this phenomenon seems not well documented yet. Therefore, it is a necessity that models incorporate this potential domain shift in the future to contribute to applications of crop type mapping in practice. To summarize, our

dataset presents two main challenges to the community: First, design new architectures which can effectively use spatial and temporal information for crop type mapping at scale. Second, ensure that models manage to generalize out of the year they have been trained on to make them applicable in real-world settings.

While we believe DENETHOR presents a significant step towards phenological monitoring near real-time, it has two main limitations. First, the area covered by our dataset is comparably small because we prioritize a high spatial and temporal resolution. Second, crop type datasets often lack geographic diversity because of label availability and class compatibility issues and our dataset is no exception. This limits the ability of developed algorithms to generalize to different geographies. Although resource-intense initiatives begin to tackle this problem [39], this remains an obstacle with ample potential for future dataset work. One option for users interested in spatial generalization is to combine our dataset with a corpus of similar built for South Africa which is linked on our GitHub repository.

## 7   Conclusion

In this paper, we present **DENETHOR**: The **D**ynamic**E**arth**NET** dataset for **H**armonized, inter-**O**perable, analysis-**R**eady, daily crop monitoring from space. It is based on daily, analysis-ready Planet Fusion data in combination with Sentinel-1 and Sentinel-2 imagery. Ground truth of crop fields and types is taken from a public registry of farmer reports. We deliberately take the test data from a different year to ensure models that incorporate this temporal shift. Our experiments underline that the effects of this shift can be large and reduce performance by around 12 percentage points in accuracy.

Additionally, we point out that exploiting temporal, spatial and spectral information at scale is not a trivial task with current methods in crop type mapping. Tests with an off-the-shelf spatial encoder in combination with widely used temporal models fall short of simple pixel averages across a field with the same temporal model. Further, the best deep learning models tested barely outperform a random forest baseline of manually curated spectral features from S1 and S2 time series. Therefore, our dataset presents a challenging task to the machine learning community that may require the design of novel methods to push the frontiers of crop type mapping with next-generation EO data.

## Acknowledgments and Disclosure of Funding

This work is jointly supported by the Helmholtz Association through the joint research school "Munich School for Data Science - MUDS", the framework of Helmholtz AI [grant number: ZT-I-PF-5-01] - Local Unit "Munich Unit @Aeronautics, Space and Transport (MASTr) and the German Federal Ministry for Economic Affairs and Energy (BMWi) under the grant DynamicEarthNet (grant number: 50EE2005).

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
