# OpenReview forum: "DENETHOR: The DynamicEarthNET dataset for Harmonized, inter-Operable, analysis-Ready, daily crop monitoring from space"
_NeurIPS.cc/2021/Track/Datasets_and_Benchmarks/Round2 — NeurIPS 2021 Datasets and Benchmarks Track (Round 2)_

### Official Review · Reviewer_vLg7 · 2021-09-19
**An interesting daily remote sensing data for crop type classification**

**Rating:** 7
**Confidence:** 4
**Correctness:** yes
**Clarity:** yes

**Strengths:**

1. The data is not only geographically precise but also of high quality.
2. The experiment is sufficient.

**Weaknesses:**

The layout and format of the paper still need to be improved

**Additional Feedback:**

no

**Documentation:**

yes

**Relation To Prior Work:**

yes

**Summary And Contributions:**

the author introduced a daily remote sensing data for crop type classification: the DynamicEarthNET dataset for Harmonized, inter-Operabel, analysis-Ready, daily crop monitoring from space.

---

> ### Author Response · Authors · 2021-09-24
> **Response**
>
> We are glad to hear that you see value in our dataset and the provided experiments. Could you elaborate on potential improvements to the layout of the paper? What exactly do you have in mind?

---

### Official Review · Reviewer_3sTq · 2021-09-21
**A dataset for monitoring crop**

**Rating:** 6
**Confidence:** 3
**Clarity:** Yes.

**Strengths:**

1. The paper is well motivated and well written. The authors demonstrate the usefulness of the temporal dimension for crop type classification.
2.  The paper provides a daily remote sensing dataset, which could facilitate research on crop type classification.

**Weaknesses:**

1. The dataset is mainly in Northern Germany and is not geographically diverse. Thus, the results and conclusions might not be generalizable to other countries.
2. The dataset documentation on the website needs to be improved.



**Additional Feedback:**

It would be great to expand the dataset to include data from different parts of the world.

**Correctness:**

The dataset is constructed in a sound way. The experiment design looks reasonable.

**Documentation:**

The dataset documentation on the website needs to be improved. More details on the dataset (e.g., data shape, data type) need to be added to the website.

**Ethics:**

No.

**Relation To Prior Work:**

Yes.

**Summary And Contributions:**

This paper introduces a dataset that includes daily remote sensing data for crop monitoring.
The dataset contains daily Planet Fusion data, Sentinel-1 radar, and Sentinel-2 optical time series for crop type classification in Northern Germany. The proposed dataset could be useful for developing novel machine learning approaches for crop type monitoring.

---

> ### Author Response · Authors · 2021-09-24
> **Response**
>
> Thank you for your feedback and we are happy to hear that you feel like our dataset could be useful.
>
> > The dataset is mainly in Northern Germany and is not geographically diverse. Thus, the results and conclusions might not be generalizable to other countries.
>
> We are mindful about this limitation which unfortunately is largely driven by reference data availability and the temporal and spatial resolution of our data. We comment on this with other reviewers as well as this limitation is discussed by several reviewers. To allow the community to assess the generalizability of methods trained on our dataset, we are currently involved in the preparation of a similar dataset with reference data from South Africa. This dataset is planned to be released in October and allows researchers a case study of generalization to another region with a different crop distribution and climate. This will, however, not alleviate the fundamental trade-off between temporal resolution and spatial coverage. We focus on the former in this work and are happy to see that other contributions in this venue focus on the latter.
>
> > The dataset documentation on the website needs to be improved.
>
> We have revised the GitHub [2] repository substantially and now provide a .gif of the time-series as illustration along with details on the dataset collection and files. We are looking forward to your feedback if this fulfills what you had in mind.
> [2]: https://github.com/lukaskondmann/DENETHOR

---

### Official Review · Reviewer_Eejx · 2021-09-22
**Useful and novel real-world remote-sensing dataset that also sheds light on inadequacies in current learning algorithms**

**Rating:** 6
**Confidence:** 4
**Clarity:** Yes, the paper is clear and well writ…

**Strengths:**

- The dataset is analysis-ready and also includes S1 and S2 inputs to the train and test tiles allowing multi-modal experiments. It is the first "daily" dataset in remote sensing enabling new methods for crop detection in greater temporal density.
- The dataset is of high quality (ARD, QA) and in compliance with EU and regulatory bodies.
- Dataset does not include field boundaries making it a useful challenge in Arable land classification that could lead to works that effectively segment crops directly.
- Logically sound baseline experiments and comparison with deep and classical ML methods. Authors show that conventional methods can still surpass deep methods (except with pixel-set encoding and self-attention). But, they also state that "current methods do not fully exploit temporal data at scale".


**Weaknesses:**

- A general tone of the work has been that new challenges (Eg: class imbalance, domain shift, temporal shift, etc) are to be tackled by new (to be investigated) methods rather than by modifying the data collection or by suggesting specific solutions/ideas.
- Collection of only 4 spectral bands (RGB+NIR) as compared with Sentinel (12 bands).
- Can the authors discuss how training models on their dataset can enable generalizations to crops grown in other regions of the world especially when the spectral signatures captured will vary, and this dataset captures only 4 bands leading to a lower scope in indices-based analysis.

**Additional Feedback:**

Minor edit:
- L8: Operabel -> operable

**Correctness:**

Yes, the submission is a dataset constructed in a manner useful for the remote sensing and machine learning communities. The evaluation methods and experiments provide insights on limitations in current machine learning approaches and show that new directions of research are necessary. The dataset and the code were shared.

**Documentation:**

Yes, the dataset collection was part of the Common Agricultural Policy of the European Union wherein farmers self-reported the crops (ground truth). Further, the quality of the data was verifiable (ARD, QA). The authors promise to release the dataset under CC-BY license making is useful for the community.


**Ethics:**

There seem to be no ethical concerns that warrant further discussions.

**Relation To Prior Work:**

Yes, this work suitably discusses prior art and also explains limitations in previous methodologies that make this contribution useful for the community. Compared with datasets in Table 1, the size is understandably higher, and the dataset covers 4,500 fields in 3m spatial resolution from Northern Germany.

**Summary And Contributions:**

- Introduce and make available a novel, analysis-ready dataset (de-clouded and harmonized) of daily (time-series) remote sensing data for crop type classification in addition to S1 and S2 for 4,500 fields at 3m spatial resolution.
- Discuss new challenges that need to be tackled in order to effectively utilize the proposed dataset. This includes domain-shift, arable land classification without field boundaries, tackling class imbalance, and utilizing increased temporal density.
- Baseline experiments investigate domain shift due to temporal variation (out-of-year-generalization).
- Suitably describes limitations of this work (and the niche for this application in general).

---

> ### Author Response · Authors · 2021-09-24
> **Response**
>
> We thank the reviewer for their time and feedback to our paper. In the following, please find our individual responses:
>
> > A general tone of the work has been that new challenges (Eg: class imbalance, domain shift,  temporal shift, etc) are to be tackled by new (to be investigated) methods rather than by modifying the data collection or by suggesting specific solutions/ideas.
>
> Of course, new and better data is equally important and in presenting this challenge we want to emphasize that we believe there is potential in methods which better data can help us identify. This by no means is supposed to express that there is no or little potential in the data itself. One reason for publishing this dataset is also to collect feedback on the data itself and analyze what “better” images can contribute to crop monitoring.
> The primary goal is the presentation of the dataset in this work. We do give some first pointers where we believe potential improvements could be found such as pixel-set encoding with temporal convolutions but this is not our focus in the paper. This is because we encourage specific solutions and ideas with the challenge that we are hosting around this dataset (4th October - 19th December) where we hope to explore first steps to improve methods together with the community.
>
> > Collection of only 4 spectral bands (RGB+NIR) as compared with Sentinel (12 bands).
>
> The dataset provides Sentinel 2 imagery on top of the usually not available Planet imagery. We would like to emphasize that also the Sentinel 2 images have only 4 bands at 10m resolution. Other bands (6x20m,3x60m) are usually upsampled to 10m but do not contain high-frequency information. A physical constraint on photons at certain wavelengths being detected within one spectral band leads to a trade-off between spatial and spectral resolutions. Planetscope satellites prioritize spatial resolution at 3m but collect only 4 bands. Sentinel-2 collects more spectral signatures but loses spatial resolution (10m-60m). This makes it difficult to achieve similar spectral signatures at different resolutions. That is why we include both data sources to achieve spatial, temporal and spectral depth.
>
> > Can the authors discuss how training models on their dataset can enable generalizations to crops grown in other regions of the world especially when the spectral signatures captured will vary, and this dataset captures only 4 bands leading to a lower scope in indices-based analysis.
>
> The ability to test geographic generalization is one limitation of our dataset because we prioritize temporal resolution over geographic extent in a region where high-quality reference data is available. We are, however, currently part of a collaboration to publish a dataset with a similar corpus based in South Africa which is planned to be released in October. This would allow a number of follow-up experiments in a different climate where also transferring knowledge from one location to the other with our data could be explored by the community. The different modalities (S2 spectrum, S1 radar data) beyond the 4 channels in Planet fusion of our data may be a tool that helps to achieve better generalization results but we acknowledge that this is a limitation of our work and also point this out in the paper.
>
> > L8: Operabel -> operable
>
> Edited, thank you for brining this to our attention

---

### Official Review · Reviewer_Kd6A · 2021-09-23
**A new dataset for daily crop monitoring from space.**

**Rating:** 7
**Confidence:** 3
**Correctness:** Yes
**Clarity:** Yes

**Strengths:**

This dataset may have good and useful social implications. It provides the research community with high quality remote sensing data, which is typically not free, to develop algorithms for monitoring crop.

**Weaknesses:**

One of the main contributions of this paper is that it suggests that we may need new approaches to fully exploit the data. But I do not think the experiments provide a strong support for it. The authors evaluate existing baselines on the new dataset and show that they only slightly outperform random forest. It would be better if the authors can also compare existing baselines with random forest on existing dataset. Without such comparisons, it is hard to tell whether this dataset is really more challenging and warrants new approaches. Furthermore, one of the baselines achieves an accuracy of 67.25% on the test set which seems to be quite high already, considering that the model is small as it can be trained with a single mid-tier GPU from a few years ago.

It would be great if the authors can provide more details on the data, such as the cost of the data if they are allowed to disclose such information. This would be beneficial to the community if they find the data useful and would like to get data upon it.

**Additional Feedback:**

Not applicable

**Documentation:**

Yes

**Relation To Prior Work:**

Yes

**Summary And Contributions:**

The authors propose a dataset that provides daily and high resolution analysis ready remote sensing data for crop type monitoring. They collect the data through a commercial provide of high-resolution satellite imagery which is not typically freely available. They show that different deep-learning baselines do not perform well on this new data, suggesting it may require new methods to fully exploit the new data.

---

> ### Author Response · Authors · 2021-09-24
> **Response**
>
> We thank the reviewer for their insightful suggestions. Please find our responses below:
>
> >It would be better if the authors can also compare existing baselines with random forest on existing dataset
>
> We see in other datasets with these temporal models that deep learning typically has the edge over shallow models in crop type mapping. For example, on the Breizhcrops dataset [1], all 4 temporal models we benchmark are similar or better in accuracy than random forests with a spatial pixel average:
> - 0.78 (RF)
> - 0.79 (TempCNN)
> - 0.77 (MS-ResNet)
> - 0.80 (LSTM)
> - 0.80 (Transformer)
>
> see [1, Table 2, L2A results]. This is mostly not the case in our paper because random forests are better than most deep learning models we evaluated. One possible explanation is that the methods designed based on S2 data with different temporal and spatial lead to suboptimal results on the new modality of daily Planet images. However, this comparison has some limitations because the Breizhcrops results are based on Sentinel-2 data with a larger geographical scale but in the same year. This is why we have not yet included this in the manuscript yet but we would be happy to adjust this if helps address the reviewer's concern. As we also mention in the paper, a larger geographic scale may also help.
>
> > Furthermore, one of the baselines achieves an accuracy of 67.25% on the test set which seems to be quite high already, considering that the model is small as it can be trained with a single mid-tier GPU from a few years ago.
>
> We believe that there is still much potential as about ⅔ accuracy is comparably low for crop type mapping. Other models within the same year and comparable regions can achieve 80% accuracy (see BreizhCrops results above) and higher which suggests there is still potential in this task.
>
> > It would be great if the authors can provide more details on the data, such as the cost of the data if they are allowed to disclose such information. This would be beneficial to the community if they find the data useful and would like to get data upon it
>
> Unfortunately, we can not disclose pricing details of Planet products publicly, as it is still an early-stage product. However, we can elaborate on the availability of the data: Planet fusion data can be produced on-demand anywhere in the world from January 2018 on. There exist several exclusive possibilities for academic institutions to obtain Fusion data such as an R&D engagement with Planet or through their education and research program
>
> [1]: Rußwurm, M., Pelletier, C., Zollner, M., Lefèvre, S., & Körner, M. (2019). BreizhCrops: A time series dataset for crop type mapping. arXiv preprint arXiv:1905.11893..

---

### Official Review · Reviewer_ow3L · 2021-09-23
**Valuable Contribution to ML for Remote Sensing Community**

**Rating:** 7
**Confidence:** 4

**Strengths:**

**S1) First publicly available benchmark dataset with daily, analysis-ready remote sensing data**

Especially since Planet Fusion imagery is typically not free for public use, DENETHOR provides researchers in the machine learning and remote sensing communities access to the latest generation of satellite imagery for analysis. It is reasonable to expect that such imagery becomes more common in the future, so developing models now can help us take advantage of the future imagery.

**S2) Positive societal implications**

Crop type mapping is critical for many use cases, including food security estimation.

**S3) Provides ML community with research tasks**

DENETHOR encourages development of machine learning methods that can handle:
- multi-modal data (Planet Fusion + Sentinel-1 + Sentinel-2)
- geospatio-temporal data
- domain shift (geospatial and temporal)
- noisy input data (clouds in Sentinel-2 imagery)

**Weaknesses:**

**W1) Difficulty untangling spatial vs. temporal generalization**

The training and testing data differ both in location and in year. This makes it difficult to disentangle the drop in generalization performance due to spatial differences vs. temporal differences. If data for both 24km-by-24km tiles could be provided for both years (2018 and 2019), then it would be easier to perform ablation studies.

**W2) Limited geographic coverage**

The dataset only includes crop data from two 24-by-24km regions in Germany, and the training set only includes one such region. Therefore, it is difficult to expect models trained on DENETHOR to perform well in other geographic regions. Especially given that crop type mapping is most critical in developing countries where data is scarce, the limited geographic coverage limits the utility of models trained on DENETHOR.



**Additional Feedback:**

N/A

**Clarity:**

No major clarity issues identified. However, the paper seems to use a large amount of remote-sensing jargon, which the broader machine learning community may be less familiar with.

**Correctness:**

The benchmark seems well-designed. The authors took care to ensure no overlap between the training and test sets. Code for reproducing baseline models is available on GitHub.

**Documentation:**

The documentation looks complete.

**Ethics:**

No concerns.

**Relation To Prior Work:**

R1) As mentioned in (S1), this is the first work to publicly and freely release daily analysis-ready satellite imagery.

R2) While the paper discusses some specific benchmark datasets for crop type classification, the paper does not seem to highlight other sources of crop type classification data, such as the US Cropland Data Layer (CDL). Although the CDL is not necessarily "benchmark-ready,"  a discussion of how DENETHOR is different from the CDL and related datasets would be appreciated.

**Summary And Contributions:**

The DENETHOR dataset provides satellite imagery from Planet Fusion (3m/px, daily revisit, RGB + near-infrared), Sentinel-1 (10m/px, 6-day revist, radar), Sentinel-2 (10m/px, 5-day revisit, 12 spectral bands) for two non-overlapping 24km-by-24km regions in Germany. Labels include crop type (9 classes) for 4,500 crop fields, thresholded at at least 1000 square meters in size. Data for the first region are from 2018, while data for the second region are from 2019.

Notably, DENETHOR is the first publicly available benchmark dataset with daily, analysis-ready remote sensing data, since Planet Fusion imagery is typically not free for public use. Experimental results on crop type classification suggest the need to develop new machine learning techniques to take advantage of the high temporal resolution, multi-modal data and improve cross-domain generalization.

---

> ### Author Response · Authors · 2021-09-24
> **Response**
>
> We thank the reviewer for their in-depth comments. We respond to the provided suggestions individually below:
>
> > W1) Difficulty untangling spatial vs. temporal generalization
>
> We appreciate the interest of the reviewer in our ablation studies which try to disentangle the effect of temporal and spatial differences. We were initially hesitant to publish both years for training and test tiles because this essentially doubles the dataset size to about 500 GB. But we agree that having both years allows users of the dataset to perform ablation studies during methodological design as well and we see the value in this. We can incorporate this as an additional download option which will not affect the accessibility of the core dataset. This will, however, only be possible with the full release of the dataset (including test data) after the competition we’re organizing with the dataset running from 4th October - 19th December 2021. We have added a sentence about this in the updated version of the paper.
>
> > W2) Limited geographic coverage
>
> As we also mention in the paper, geographic coverage of one region in Germany is a limitation and the consequence of our focus on high quality data we could gather in this region. This is one effect of providing the full-time series of imagery not yet cropped to the fields. We believe providing the full images is crucial for potential future experiments with our datasets such as declouding. To alleviate this limitation, we are part of a collaboration that builds a similar dataset for South Africa in cooperation with local authorities. This will allow follow-up experiments about spatial generalization and focus more on food security in low- and middle-income countries. This other dataset is planned to be released in October of this year and we will reference the opportunity for further experiments on the repository once available
>
> > The paper seems to use a large amount of remote-sensing jargon, which the broader machine learning community may be less familiar with.
>
> We have given our paper to colleagues from the ML community without much remote sensing exposure for a designated jargon review. This has returned some minor points of potential improvement, particularly in the satellite data section. For example, we have added a short explanation of orthorectification in the updated manuscript. We are still waiting to hear back from a second jargon reviewer and will update the paper accordingly
>
> > While the paper discusses some specific benchmark datasets for crop type classification, the paper does not seem to highlight other sources of crop type classification data, such as the US Cropland Data Layer (CDL)
>
> We have added a reference to CDL in the related work section: “The rarity of datasets may primarily be a result of the low availability of high-quality reference data at scale. Large-scale products of crop types such as the Cropland Data Layer (CDL) [4] in the US exist for some regions. However, the CDL is technically still a prediction and may not provide sufficient quality for benchmark purposes.”
>
> from references:
> [4] Boryan, C., Yang, Z., Mueller, R., & Craig, M. (2011). Monitoring US agriculture: the US department of agriculture, national agricultural statistics service, cropland data layer program. Geocarto International, 26(5), 341-358.

---

> > ### Comment · Reviewer_ow3L · 2021-10-03
> > **Concerns addressed**
> >
> > Thank you to the authors for addressing my main concerns. I am keeping my score as-is, with the good-faith understanding that the authors will further address the points below for their camera-ready submission.
> >
> > > W1) Difficulty untangling spatial vs. temporal generalization
> > >
> > > W2) Limited geographic coverage
> >
> > I'm trusting that the authors will follow through on their claim that they will release both years of the dataset covering the same regions in Germany, as well as provide the South Africa dataset. Hopefully, the South Africa dataset can also be described in the camera-ready version of the paper, given that the deadline for the camera-ready is in November.
> >
> > > Remote sensing jargon
> >
> > I appreciate the efforts made to explain orthorectification. However, more terms that should be explained include:
> >
> > - "analysis-ready data (ARD), preprocessing level 3 product"
> > - "the Fusion data is calibrated to the HLS spectrum which eliminates differences in the spectral signatures"
> > - "the vertical-vertical (VV) and vertical-horizontal (VH) polarisations" - nowhere else in the paper is polarisation discussed. I know that Sentinel-1 uses radar (which therefore has polarisation), but a general ML audience is unlikely to be familiar with this background knowledge.
> >
> > **Additional Clarity Concerns**
> >
> > In Section 4, under "Deep learning models," the paper describes the generic encoders, involving constants $H, C, T$. Please specify the concrete values you used for $H,C,T$ in your actual baseline models.

---

### Decision · Program_Chairs · 2021-10-10

**Decision:**

Accept

**Comment:**

All reviewers recommended accept. AC finds no grounds to overturn the consensus.